# Influence of Certification Program on Treatment Quality and Survival for Rectal Cancer Patients in Germany: Results of 13 Certified Centers in Collaboration with AN Institute

**DOI:** 10.3390/cancers16081496

**Published:** 2024-04-13

**Authors:** Mihailo Andric, Jessica Stockheim, Mirhasan Rahimli, Sara Al-Madhi, Sara Acciuffi, Maximilian Dölling, Roland Siegfried Croner, Aristotelis Perrakis

**Affiliations:** Department of General, Visceral, Vascular and Transplant Surgery, University Hospital Magdeburg, Leipziger Str. 44, 39120 Magdeburg, Germany; mihailo.andric@med.ovgu.de (M.A.); jessica.stockheim@med.ovgu.de (J.S.); mirhasan.rahimli@med.ovgu.de (M.R.); sara.al-madhi@med.ovgu.de (S.A.-M.); sara.acciuffi@med.ovgu.de (S.A.); maximilian.doelling@med.ovgu.de (M.D.); roland.croner@med.ovgu.de (R.S.C.)

**Keywords:** rectal cancer, certification, colorectal cancer center, outcome

## Abstract

**Simple Summary:**

In the past, the German Cancer Society has implemented a certification program for colorectal cancer centers with the aims of standardizing oncological treatment, endorsing a multidisciplinary approach, and improving the outcomes. However, some critical views have argued that fulfilling the certification requirements alone would not necessarily enhance the treatment quality for colorectal cancer patients. In the present study, our objective was to investigate the treatment outcomes for patients with rectal cancer in hospitals of different medical care levels, before and after the certification process. The results of the present study indicate an improvement in terms of the treatment quality and outcomes after the official certification process. Further prospective clinical trials are necessary to investigate the influence of certification on the treatment of patients suffering from colorectal cancer.

**Abstract:**

Introduction: The certification of oncological units as colorectal cancer centers (CrCCs) has been proposed to standardize oncological treatment and improve the outcomes for patients with colorectal cancer (CRC). The proportion of patients with CRC in Germany that are treated by a certified center is around 53%. Lately, the effect of certification on the treatment outcomes has been critically discussed. Aim: Our aim was to investigate the treatment outcomes in patients with rectal carcinoma at certified CrCCs, in German hospitals of different medical care levels. Methods: We performed a retrospective analysis of a prospective, multicentric database (AN Institute) of adult patients who underwent surgery for rectal carcinoma between 2002 and 2016. We included 563 patients from 13 hospitals of different medical care levels (basic, priority, and maximal care) over periods of 5 years before and after certification. Results: The certified CrCCs showed a significant increase in the use of laparoscopic approach for rectal cancer surgery (5% vs. 55%, *p* < 0.001). However, we observed a significantly prolonged mean duration of surgery in certified CrCCs (161 Min. vs. 192 Min., *p* < 0.001). The overall morbidity did not improve (32% vs. 38%, *p* = 0.174), but the appearance of postoperative stool fistulas decreased significantly in certified CrCCs (2% vs. 0%, *p* = 0.036). Concerning the overall in-hospital mortality, we registered a positive trend in certified centers during the five-year period after the certification (5% vs. 3%, *p* = 0.190). The length of preoperative hospitalization (preop. LOS) was shortened significantly (4.71 vs. 4.13 days, *p* < 0.001), while the overall length of in-hospital stays was also shorter in certified CrCCs (20.32 vs. 19.54 days, *p* = 0.065). We registered a clear advantage in detailed, high-quality histopathological examinations regarding the N, L, V, and M.E.R.C.U.R.Y. statuses. In the performed subgroup analysis, a significantly longer overall survival after certification was registered for maximal medical care units (*p* = 0.029) and in patients with UICC stage IV disease (*p* = 0.041). In patients with UICC stage III disease, we registered a slightly non-significant improvement in the disease-free survival (UICC III: *p* = 0.050). Conclusions: The results of the present study indicate an improvement in terms of the treatment quality and outcomes in certified CrCCs, which is enforced by certification-specific aspects such as a more differentiated surgical approach, a lower rate of certain postoperative complications, and a multidisciplinary approach. Further prospective clinical trials are necessary to investigate the influence of certification in the treatment of CRC patients.

## 1. Introduction

With over 60,000 new cases and over 25,000 deaths annually, colorectal cancer is still one of the most common malignant diseases in Germany [1]. Up to 38% of these patients suffer from cancer of the rectum [1,2,3,4]. Over the last 20 years, rectal cancer treatment has evolved, and nowadays, it involves a multidisciplinary approach that includes standardized diagnostics, neoadjuvant and adjuvant therapy (if these are indicated), and interventional and supportive treatment modalities. In this setting, and in cases of a curative intent, surgery plays the most important role [5,6,7,8,9,10,11]. In order to standardize oncological treatment and improve the outcomes in colorectal cancer (CRC) patients, the centralization of treatment in specialized high-volume centers and the certification of oncological units as colorectal cancer centers (CrCCs) have been proposed in Germany [12].

This is a part of the certification program of the German Cancer Society, which was developed in Germany and has expanded to other member states of the European Union. The certification program for breast cancer was introduced in 2003 and the program for colorectal cancer was introduced in 2006, later being applied to centers for malignancies of diverse organ systems. The goal was to offer a treatment that is based on high quality standards at every stage of the disease [11]. Certified cancer centers form the base of this approach. The centers are required to annually demonstrate their outcomes and are obliged to meet the technical and medical requirements for the treatment of a specific tumor entity [11]. Medical guidelines (in the case of CrCCs, the “S3 guidelines for the treatment of colorectal cancer”) represent the foundation for defining these quality standards [6].

However, it is not obligatory for oncological units to undergo the certification program in order to be allowed to treat specific malignances. In 2017, only 47.15% of the overall patients with colorectal cancer in Germany were treated in a CrCC, and in 2018, 53% were treated in a certified colorectal cancer center [1,12,13].

Benz et al. described an increasing proportion of rectal cancers treated in certified centers, rising from 43% to 57% during the period from 2010 to 2018 [13]. Hence, 43% of the overall rectal cancer cases in Germany are being treated in uncertified centers, with a case load of <20 operative cases per year [13].

Several recent studies have demonstrated some advantages when treatments take place in certified centers, such as a better overall survival and a lower morbidity, especially for advanced colorectal cancer patients [3,5,14,15,16]. On the other hand, since the introduction of the German certification program, critical views have argued that fulfilling the certification requirements alone would not necessarily enhance the treatment quality for colorectal cancer [17]. The achieved effects of centralization for CRC treatments have been described as insufficient, and a renewal of national strategies with a focus on the implementation of centralization and high-quality CrCCs was proposed [2].

Therefore, we aimed to investigate the influence of colorectal cancer center (CrCC) certification on the treatment outcomes for rectal carcinoma patients according to the database of the AN Institute of the Otto von Guericke University of Magdeburg. We evaluated data from 13 hospitals of different medical care levels on rectal cancer treatments in Germany for the period of 2002–2016.

## 2. Methods

### 2.1. Study Design

A comparative, retrospective study was conducted using data from a prospectively acquired, multicenter database of the AN Institute of the Otto von Guericke University in Magdeburg, Germany. All of the data were acquired from the 13 associated hospitals based on standardized documentation forms, which were drafted by the scientific advisory board of the AN Institute. Since 2010 and after the implementation of the certification process by the German Cancer Society, the scientific advisory board of the AN Institute revised all of the documentation for tumor entities according to the high standards defined by the German Cancer Society.

A total of 563 patients from 13 hospitals that received treatment from five years before until five years after the official certification of the center as a CrCC (ten-year period for each observed center) were examined. The patients treated during the five-year period before the certification of a particular center were included in the group defined as “−5y” and the patients treated during the five-year period after the certification were included in the group defined as “+5y” (Figure 1). Overall, we included patients that were surgically treated during the period of 2002–2016, meaning that the certifications of all of the included centers took place between the years 2007 and 2011.

We performed a comparative analysis of the patient characteristics, perioperative parameters, postoperative outcomes (including morbidity and mortality), and survival data between the selected collectives from the period before and the period after the date of official certification as a colorectal cancer center (−5y vs. +5y). A subgroup analysis of survival according to the hospital care level and the UICC stage was also performed. 

### 2.2. Inclusion Criteria

In the present study, we included adult patients (>18 years old) who underwent surgical treatment for rectal carcinoma. The patients treated during the period from 2002 to 2016 were included in the study (n = 563).

### 2.3. Exclusion Criteria

Patients <18 years old and patients who did not receive surgery for rectal carcinoma were excluded from the study.

### 2.4. Statistical Analysis

For the statistical analyses, we used SPSS 26, SPSS Inc., IBM, Armonk, New York, NY, USA.

For the data presentation, we used the means with the standard deviation or the number of cases with percentages in accordance with the type of data. The analysis and visualization of survival data were performed using a Kaplan–Meier curve. *p*-values of <0.05 were considered statistically significant.

## 3. Results

### 3.1. Patient Characteristics

All of the data related to demographic characteristics are presented in Table 1.

A total of 563 patients were included in the study, with 267 of the patients treated during the five-year period before certification (−5y) and 296 of the patients treated after certification (+5y).

There was no significant difference between the −5y and +5y groups in the proportion of patients belonging to each sex, the mean age of the patients, the mean BMI of the patients, or the distribution of cases according to the American Society of Anesthesiologists (ASA) score.

### 3.2. Perioperative Parameters

All of the data related to the perioperative parameters are presented in Table 2.

The frequency of a minimally invasive approach (laparoscopy) for rectal surgery increased significantly after CrCC certification in comparison to the period before certification (5% vs. 55%, *p* < 0.001), without a significant increase in the conversion rate. Furthermore, the frequency of a trans-anal approach for the local excision of rectal carcinoma significantly decreased after the certification for the compared periods (8% vs. 1%, *p* < 0.001).

Regarding the type of performed surgery for rectal carcinoma, the certification process did not result in any significant changes in the proportion of anterior rectal resections (ARRs), low anterior rectal resections (LARRs), or abdominoperineal extirpations (APEs) performed between the examined periods.

On the other hand, discontinuous resections according to Hartmann increased significantly during the five years after certification (+5y) in comparison to the period before certification (−5y) (1% vs. 7%, *p* < 0.001).

The mean duration of surgery for rectal cancer showed a significant increase during the five-year period after certification (161 Min. vs. 192 Min., *p* < 0.001).

### 3.3. Postoperative Parameters

According to the performed analysis, the mean preoperative in-hospital stay length (preop. LOS) prior to a rectal surgery, which was mainly for completing the diagnostics, decreased significantly after certification (4.71 days vs. 4.13 days, *p* < 0.001).

On the other hand, the postoperative in-hospital stay duration (postop. LOS) shortened non-significantly after certification (16.65 days vs. 15.15 days, *p* = 0.151).

The overall length of stay (oLOS) also did not significantly change after CrCC certification (20.32 days vs. 19.54 days, *p* = 0.065), although a tendential shortening of the oLOS was observed.

Regarding the type of case dismissal following a rectal surgery (discharge, transfer to other units (such as rehab, neurology, nephrology, etc.), or death), we observed a tendency in the five-year period after certification towards more successful patient discharges (89% vs. 94%) and fewer transfers to other units (6% vs. 3%) and postoperative death cases (5% vs. 3%). Still, there was no statistical significance for this observation (*p* = 0.060).

The overall proportion of postoperative complications did not significantly change after the CrCC certification for the examined periods (23% vs. 27%, *p* = 0.284). However, the analysis of specific surgical complications showed an increase in postoperative intestinal atonia (for over 3 days) during the five-year period after certification (2% vs. 6%, *p* = 0.025) and a decrease in the occurrence of postoperative stool fistulas (small or large bowel, other than the anastomotic region; 2% vs. 0%, *p* = 0.036). Other specific surgical complications underwent no significant change after the certification process, as displayed in Table 3.

The rate of non-surgical complications did not change significantly after CrCC certification (20% vs. 18%, *p* = 0.552). An analysis of specific non-surgical complications, such as urinary infections, non-infectious pulmonal complications, pneumonia, cardiac complications, thrombosis, lung artery embolisms, renal failure, and multiple organ failure, showed no significant changes between the two collectives.

### 3.4. Histopathology

The histopathological findings for the examined periods are presented in Table 4.

The number of patients with histologically verified rectal cancer prior to surgery increased significantly during the five-year period after the certification of a hospital as a colorectal cancer center (84% vs. 93%, *p* = 0.001).

Concerning the distribution of pN stages for the patients in both collectives, we registered a significant difference when the +5y period was compared to the reference period (−5y) (*p* = 0.001). Additionally, the frequency of an unclear lymph node status after the histopathological findings (pNx) was significantly reduced after certification (8% vs. 2%).

We observed a significant difference in the L status before and after certification, with a decreasing number of cases not being histologically examined for their L status (22% vs. 7%, *p* < 0.001). This resulted in an increased proportion of patients with an L0 or L+ status.

A significant difference in the V status was documented between the five-year periods before and after certification, with a decreasing number of cases not being histologically examined for their V status (24% vs. 8%, *p* < 0.001) and a subsequent increase in the number of patients with a V0 or V+ status.

A patient distribution analysis according to the UICC showed a significant difference in the distribution for the five-year periods before and after certification, with an obvious increase in the number of cases with UICC stage II or UICC stage III, but a decreasing number of UICC IV cases (*p* < 0.001).

Concerning the quality of the surgical treatment in terms of the M.E.R.C.U.R.Y. status and the coning of the specimen [18,19,20,21,22,23], a comparative analysis could not be performed because the data related to these parameters were only available after the certification. These particular values are presented in Table 4.

### 3.5. Follow-Up and Survival

#### 3.5.1. Survival before and after Certification for the Entire Collective and According to the Medical Care Level

Regarding the overall survival, the comparison using Kaplan–Meier curves indicated a better outcome during the five-year period after the certification (+5y) when compared to the collective treated during the five years before certification (−5y), although without reaching significant levels (*p* = 0.503).

The subgroup analysis showed a trend towards a slightly higher overall survival during the five-year period after certification for the patients from hospitals with a basic or priority level of medical care, without significance (*p* = 0.750; *p* = 0.638). A significant improvement in the overall survival was observed for the patients with rectal cancer treated in the hospitals with a maximal level of medical care during the five-year period after certification compared to the reference period (*p* = 0.029) (Figure 2).

Concerning the disease-free survival during the five-year period after the certification, the Kaplan–Meier curves revealed a non-significantly better outcome after certification for the whole collective (*p* = 0.163) as well as for the subgroups from the centers with a basic, priority, or maximal care level (*p* = 0.583; *p* = 0.845; *p* = 0.073) (Figure 3).

#### 3.5.2. Survival According to UICC Stage

##### Survival before and after Certification According to UICC Stage

The Kaplan–Meier analysis according to the UICC stage of rectal cancer showed a negative correlation between the overall survival and a higher UICC stage within the −5y group (I vs. II: *p* = 0.298; I vs. III: *p* < 0.001; I vs. IV: *p* < 0.001; II vs. III: *p* = 0.061; II vs. IV: *p* < 0.001; and III vs. IV: *p* < 0.001), as well as within the +5y group (I vs. II: *p* = 0.032; I vs. III: *p* = 0.001; I vs. IV: *p* < 0.001; II vs. III: *p* = 0.226; II vs. IV: *p* < 0.001; and III vs. IV: *p* = 0.015) (Figure 4).

The disease-free survival in the Kaplan–Meier curves, in relation to the UICC stage of rectal cancer, showed successively shorter values with an increasing UICC stage for the −5y collective (I vs. II: *p* = 0.129; I vs. III: *p* < 0.001; and II vs. III: *p* = 0.140) and for the +5y collective (I vs. II: *p* = 0.12; I vs. III: *p*< 0.001; and II vs. III: *p* = 0.275) (Figure 5*)*.

##### Survival before and after Certification According to Particular UICC Stage

In the comparison of the UICC stages experienced by the −5y collective with those experienced by the +5y collective, the analysis showed a non-significantly longer overall survival for UICC stages I and III (I: *p* = 0.347; III: *p* = 0.248), a non-significantly shorter overall survival for UICC stage II (II: *p* = 0.383), and a significantly longer overall survival for UICC stage IV (*p* = 0.041) (Figure 6).

An analysis of the disease-free survival showed a non-significant improvement during the five-year period after CrCC certification for UICC stages I, II, and III (I: *p* = 0.188; II: *p* = 0.106; and III: *p* = 0.050) (Figure 7).

#### 3.5.3. Neoadjuvant and Adjuvant Treatment

Overall, we found that for all of the patients treated in the five years after certification, neoadjuvant chemoradiation (alternatively short radiation 5 × 5 Gy) was indicated for those with advanced rectal carcinoma (cT3/cT4 and/or cN+) in the staging imaging diagnostics (computed tomography (CT) and/or magnetic resonance imaging (MRI)). For patients with a nodal-positive status in their postoperative histopathological examination, adjuvant chemotherapy was indicated, as defined by the German S3 guidelines [6]. As for the patients treated before certification, we did not have enough data for either the neoadjuvant or the adjuvant treatment, and therefore, we could not perform a comparative analysis.

## 4. Discussion

Since the implementation of a certification system for oncological units in Germany, there have been divided opinions regarding whether the treatment quality and outcomes for patients with colorectal cancer would improve by meeting the criteria of the colorectal cancer centers (CrCCs) [17].

On the other hand, several studies have shown the advantages of certified, high-volume centers and the positive influence of multidisciplinary treatments on the outcomes [5,12,24,25]. For instance, a significantly better three-year overall survival for colorectal cancer was shown for patients treated in a certified center, compared to patients treated in uncertified units (71.6% vs. 63.6%, *p* = 0.001) [24]. In another study, a significant prolongation of the relative survival was observed in UICC IV rectal cancer patients treated in an experienced, certified center compared to the national average outcomes [5]. Finally, the latest WiZen study (2023) showed a longer five-year overall survival for different tumor entities, including rectal cancer (49.2% vs. 43.3%), for patients who had received an initial treatment in a certified center compared to those treated in an uncertified center [26].

Concerning the case load of oncological units and specialized surgeons, centers with a higher volume are known to achieve better outcomes with a lower morbidity and a longer overall survival [14,15,24]. Furthermore, in the study by Ghadban et al., which included an analysis of 351,028 colorectal cancer cases, a significant improvement was observed in terms of the mortality (3.8% in 2005 vs. 3.0% in 2015; *p* < 0.001), whereas the morbidity did not improve [2].

Hence, the aim of this study was to evaluate the effects of the colorectal cancer center (CrCC) certification process on the perioperative and long-term oncological outcomes for rectal cancer patients according to the database of the AN Institute of the Otto von Guericke University of Magdeburg. The main goal of this distribution was to present the advantages offered by a certification process in a dynamic fashion and while considering a timeline. Therefore, we examined the differences among 13 centers, considering the most important timeline milestone of 5 years.

Our analysis showed an increasing proportion of laparoscopic approaches from 5% to 55% during the five-year period after certification (*p* < 0.001), without an increase in the conversion rate (*p* = 0.504). Although the certification program does not directly obligate certified centers to perform a certain proportion of minimally invasive approaches for rectal surgery, an international trend of non-inferior oncological outcomes and a reduced perioperative morbidity for laparoscopies in comparison to open approaches was observed in the examined collectives after certification [6,27,28]. The proportion of laparoscopic rectal resections until 2016 in the examined collective was even higher compared to the increase from 12.3% to 48.1% reported in the German data published by Schnitzbauer et al. [29]. However, the presented trend in our data did not reach the proportion of laparoscopic colorectal surgeries performed in England within the LAPCO-program (an increase from 44% to 66%) [30]. Data concerning robotic approaches used for rectal surgeries were not available for the current analysis, as robotic colorectal surgery was developed in the included centers after the investigated period.

Furthermore, we reported a significant reduction in the number of trans-anal excisions performed after CrCC certification (8% to 1%, *p* < 0.001). The S3 guideline clearly recommends neoadjuvant chemoradiotherapy followed by a total mesorectal excision (TME) for most cases of low rectal cancers [6], and we witnessed an expansive development of endoscopic resections for adenoma and early rectal cancers in recent years [31]; therefore, we assumed that one of the above-mentioned treatment options was indicated for a greater number of patients over the studied timeline, which resulted in a decrease in the number of local surgical excisions.

The significant prolongation of the operating time after certification could be explained due to the increase in the frequency of the laparoscopic approaches. Prolonged operating times for laparoscopic approaches compared to conventional open approaches are well known and have already been reported in several studies [27].

The reported significant increase in the frequency of discontinuous resections according to Hartmann from 1% to 7% during the five-year period after certification (*p* < 0.001) was similar to that reported by Klaue et al. [17]. This observation was interpreted in the mentioned study as a possible result of the “fear of anastomotic leakage” in emergency cases, especially during the first years of a certification program [17]. After conducting thorough research and performing a statistical analysis on our collective, we did not find any significant correlation between the ASA score, age, or comorbidities and the frequency of Hartmann’s resections. However, we observed a non-significant trend towards more emergency cases, as a possible explanation. The reported rate of Hartmann-reversal procedures might also be a sign of the adoption of a more careful and thoughtful approach, as reported elsewhere [32,33,34,35].

As a further aspect of improved management triggered by the certification process, several organizational advantages have been observed during the period after certification: the preoperative in-hospital length of stay was significantly reduced after certification. Furthermore, we observed a trend towards a reduction in the postoperative in-hospital length of stay. Additionally, there was a clear tendency towards a shorter overall in-hospital length of stay during the period after certification, which was comparable to other studies [36]. As reported by Aravani, the risks for a prolonged length of stay after colorectal cancer surgery are an older age (>80 years), socioeconomic deprivation, and the occurrence of a rectal cancer diagnosis [36]. The generally long in-hospital stays in the presented collective matched the data of other colorectal units in Germany in 2016 (18.6 ± 11.9 days), with a further shortening in the following years (13.8 ± 9.3 days in 2021) [37]. Although the time frames shown here represent the historic philosophy of perioperative management (diagnostics under stationary conditions, and long in-house stays until wound sutures were removed), which strongly varies from the current fast-track surgery goal [37,38,39], the data showed an obvious development after certification in terms of shortened in-hospital stays.

Regarding the short-term outcomes, there was a tendential shift towards an increased proportion of successful patient discharges and a decreased number of transfers to other units within the five-year period after certification. This observation could possibly be explained by the more successful and multidisciplinary handling of complicated and/or prolonged postoperative courses, involving aspects such as physiotherapy and professional nutritional support, with a reduced need to transfer patients to other specialized units. Regarding the overall in-hospital mortality, we registered a positive trend in certified centers during the five-year period after certification (decrease from 5% to 3%). These dynamics matched the significant decrease in the mortality indicated by the above-mentioned study (3.8% in 2005 vs. 3.0% in 2015; *p* < 0.001) [2].

Regarding the frequency of both surgical and non-surgical postoperative complications, we saw a significant improvement in terms of postoperative stool fistulas of the small or large bowel (other than the anastomotic region) after certification. This could possibly be an effect of the engagement of more experienced colorectal surgeons, as required by the German Cancer Society (chosen, responsible operators for the CrCC) for rectal cancer surgeries after certification.

On the other side, during the five-year period after CrCC certification, there was a significant increase in the frequency of postoperative intestinal atonia. This observation was contradictory to the results of different studies that have suggested faster postoperative bowel movements after a laparoscopic approach [40,41]. This could be the result of the standard postoperative analgetic regimen, which was mainly opioid-based in most centers associated with the AN Institute. This regimen could have led to increased intestinal atonia, as reported in several studies [42].

Histopathological findings represent a fundamental quality control after a surgical treatment of a tumor and are indispensable for further oncological treatment. Therefore, standardized histopathological reports involving all of the important tumor characteristics and the quality of the performed surgery are one of the most important requirements for the colorectal cancer center certification process [6].

Histological verification of a diagnosis is currently the gold standard before proceeding with a multidisciplinary treatment for rectal carcinoma, especially when a neoadjuvant treatment is indicated [6]. During the five-year period after certification, the number of histologically secured diagnoses before the treatment significantly increased, nearly achieving the international high-quality levels as described in the CONCORD study (94%) [3].

Although there was no relevant change in the patient distribution regarding the pathological T stage after CrCC certification, there were significant dynamics in the distribution of cases according to the pN stage within the five-year period after certification. However, the proportion of patients with an unclear postoperative lymph node status (pNx) decreased in the +5y group from 8% to 2%, indicating an increased level of engagement for full, detailed histopathological findings after certification.

Another sign of the standardization of the histopathological findings after certification was the significantly more frequent description of lymphovascular and vascular infiltration in our collective. Although the clinical relevancy of vascular infiltration (V) has not been proven, lymph vessel infiltration (L) is correlated with a higher risk of lymph node metastases [6]. Therefore, the German guideline for the treatment of colorectal cancer recommends providing a description of the L and V parameters within a TNM classification [6].

Although an analysis of surgical quality development in terms of the M.E.R.C.U.R.Y. status and the coning of surgical specimens after a total mesorectal excision (TME) could not be performed in this study, the fact that these parameters were involved in the database only after certification (+5y) showed a raised awareness of the importance of specimen quality. However, the German S3 guideline highly recommends a TME as a standard surgical technique for the treatment of rectal carcinoma, and the M.E.R.C.U.R.Y. status represents the pathological description of its quality [6,18,23]. In the study published by Sahm et al., which involved analyzing the quality of care for colorectal cancer in the federal state of Brandenburg, Germany, the reported rate of M.E.R.C.U.R.Y. I rectal resections was 96.4% in certified colorectal centers [4]. In the present study, we reported a similar rate of 94% during the five-year period after certification, suggesting at least that the recommended surgical technique was implemented after CrCC certification.

The overall survival in our collective slightly improved after certification, with a reported significance for patients with UICC stage IV disease (*p* = 0.041). This observation has also been reported by Richter et al. [5].

In terms of disease-free survival, we recorded the improvement in UICC stage III patients, only just falling short of a significant level (*p* = 0.05).

The subgroup analysis according to the medical care level (basic-, priority-, and maximal-care-level hospitals) showed a slightly superior overall survival and disease-free survival during the five-year period after certification, compared to the period before. A significant improvement in the overall survival was documented for the hospitals with a maximal level of medical care during the five-year period after certification, compared to the reference period (*p* = 0.029). We interpreted this observation as the result of a multifactorial process, including the increasing quality of the diagnostics, the selection of adequate multidisciplinary treatment modalities according to the guidelines of the German Cancer Society and the “best standard of care”, the implementation of improved surgical techniques, the increased quality of histopathological documentation, and further immeasurable aspects, all indicated by the certification process.

The strength of this study lies in its comparison of rectal cancer data between defined periods before and after certification, including the survival rate in hospitals of different care levels as well as within particular stages according to the UICC classification. This comparison offers a direct view into the possible effects of certification. As mentioned before, the main goal of this distribution was to present the advantages offered by a certification process in a dynamic fashion while considering a timeline. Therefore, we offer the differences seen among 13 centers, considering the time frame of five years before and after certification. Such a comparative analysis is missing in the current literature.

### Limitations

This study has several limitations. It is a retrospective analysis of a database that, while prospectively acquired, is still missing relevant data for the treatment of rectal cancer, such as neoadjuvant and adjuvant treatments, especially for patients treated prior to certification. The latter is, in our opinion, the most important limitation of the current study. There is heterogeneity as far as data collection and the quality of data are concerned. The main reason for this is a dynamic change in the gold standards for the treatment of rectal cancer, such as the use of neoadjuvant chemoradiation, the meaning of the M.E.R.C.U.R.Y. classification for surgical quality and its impact on recurrence rates, etc. The implementation of the certification process for colorectal cancer centers by the German Cancer Society was the breakthrough in terms of the data sampling quality and tumor documentation in Germany. The certification has driven centers to optimize their databases and raise their parameters. Therefore, a certification process is also important in terms of the quality of tumor documentation and treatments according to the guidelines. Although this was a multicentric study that included centers from different federal states, it only considered data up to 2016, and thus may not be representative of Germany as a whole and of the state of the art for rectal cancer treatment in 2023. Information about the case loads of certain hospitals was also not available.

Nevertheless, we firmly believe that a dynamic evolution process within the scope of a structured certification program cannot be managed only by statistical programs, because of the multivariability in the quantitative and qualitative aspects involved. Due to the latter aspect, a dynamic and continuous evaluation over a timeline is indispensable. A five-year milestone is the most important time milestone, as determined by the German Cancer Society.

## 5. Conclusions

The results of the present study indicate a clear trend towards improvement in terms of the treatment quality, survival, and documentation in certified colorectal cancer centers. This is demonstrated by certification-specific aspects such as more differentiated surgical approaches, a lower rate of certain complications, and a multidisciplinary approach. In our honest opinion, the qualitative aspects of a certification process (such as the need for multimodal treatment, the need to follow the guidelines and the current advancements in treatments, the need for interdisciplinary tumor boards, etc.), together with a stable volume of cases as proposed by the German Cancer Society, are the essential aspects required for improvement. Further prospective clinical trials are needed to investigate the relevance of certification in the treatment of CRC patients.

## Figures and Tables

**Figure 1 cancers-16-01496-f001:**
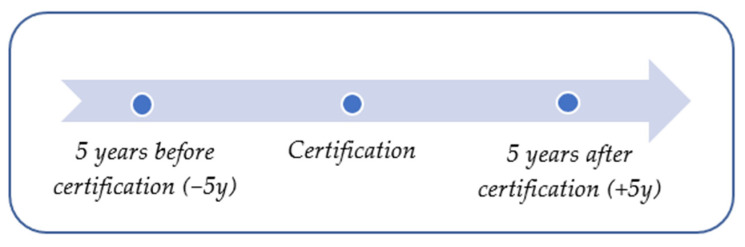
Presentation of time-related collective building according to the moment of the hospital’s certification as a colorectal cancer center.

**Figure 2 cancers-16-01496-f002:**
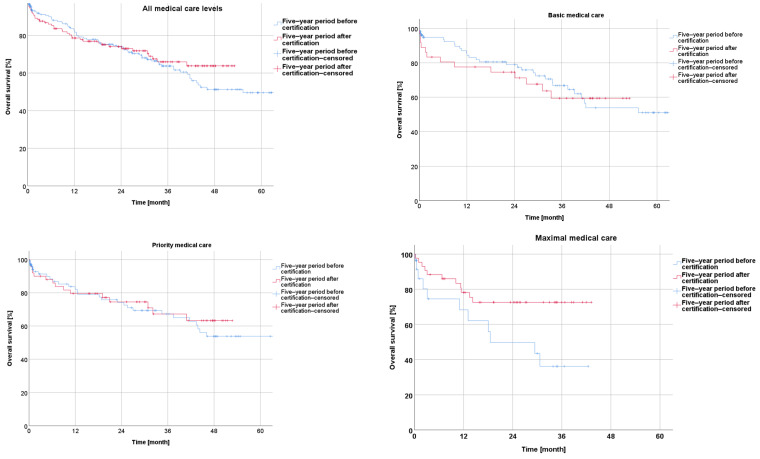
Presentation of overall survival five years before and five years after certification according to medical care level.

**Figure 3 cancers-16-01496-f003:**
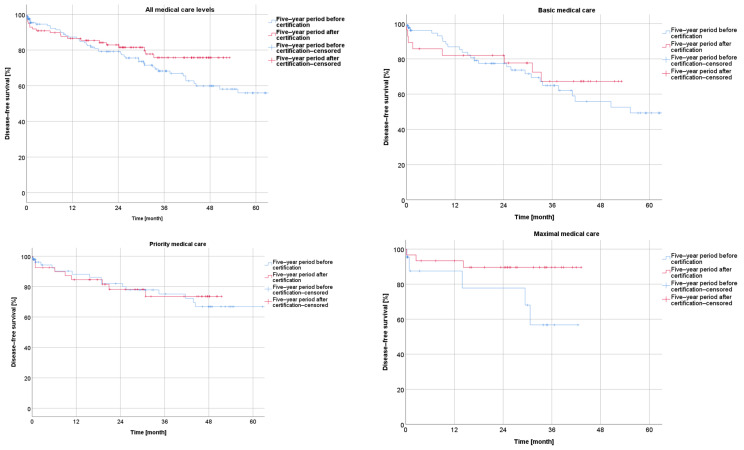
Presentation of disease-free survival five years before and five years after certification according to medical care level.

**Figure 4 cancers-16-01496-f004:**
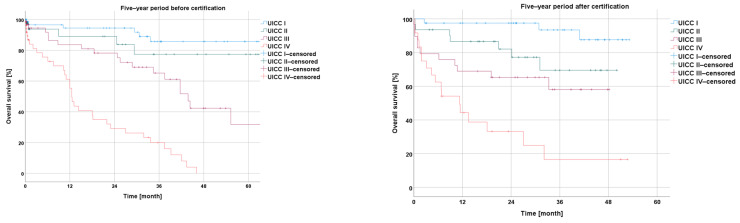
Presentation of overall survival five years before and five years after certification according to UICC stage.

**Figure 5 cancers-16-01496-f005:**
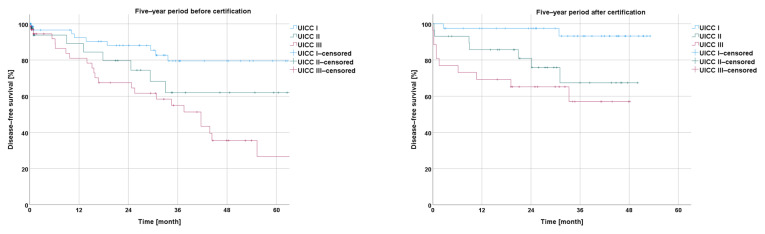
Presentation of disease-free survival five years before and five years after certification according to UICC stage.

**Figure 6 cancers-16-01496-f006:**
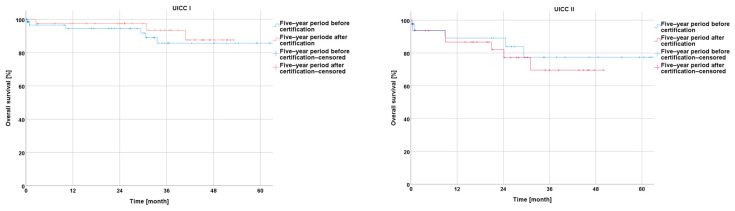
Presentation of overall survival five years before and five years after certification according to particular UICC stage.

**Figure 7 cancers-16-01496-f007:**
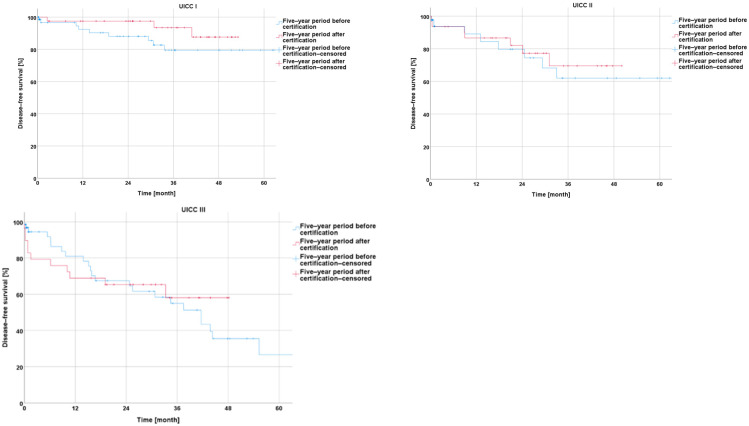
Presentation of disease-free survival five years before and five years after certification according to particular UICC stage.

**Table 1 cancers-16-01496-t001:** Presentation of patient characteristics for collectives treated five years before and five years after certification.

	CrCC Certification				
	−5y		+5y		*p*
Parameter	N (Mean)	% (SD)	N (Mean)	% (SD)	
Number of patients	267	47.42%	296	52.58%	
Sex					0.925
Female	109	41%	122	41%	
Male	158	59%	174	59%	
Age	68	10.51	68	12.67	0.712
BMI	26	4.36	27	4.83	0.467
ASA					
ASA I	24	9%	27	9%	0.234
ASA II	153	57%	145	49%	
ASA III	87	33%	120	41%	
ASA IV	3	1%	3	1%	

**Table 2 cancers-16-01496-t002:** Presentation of perioperative parameters for collectives five years before and after certification.

	CrCC Certification				
	−5y		+5y		*p*
Parameter	N (Mean)	% (SD)	N (Mean)	% (SD)	
Number of patients	267	47.42%	296	52.57%	
Surgical approach					
Laparotomy	226	86%	114	39%	<0.001
Laparoscopy	14	5%	160	55%	<0.001
Conversion	2	6%	16	9%	0.504
Trans-anal	20	8%	3	1%	<0.001
Surgery type					
ARR	47	18%	47	17%	0.753
LARR	113	43%	112	40%	0.505
APE	43	16%	62	22%	0.084
Hartmann	3	1%	23	8%	<0.001
Anastomosis type					
Stapler	167	63%	178	61%	0.572
Intraoperative complications	14	5%	15	5,17%	0.93
Duration of surgery (Min.)	161	74.21	192	79.33	<0.001

**Table 3 cancers-16-01496-t003:** Presentation of postoperative parameters for collectives five years before and five years after certification.

	CrCC Certification				
	−5y		+5y		*p*
Parameter	N (Mean)	% (SD)	N (Mean)	% (SD)	
Number of patients	267	47.42%	296	52.57%	
LOS					
Preop. LOS (days)	4.71	4.55	4.13	17.95	<0.001
Postop. LOS (days)	16.65	14.88	15.15	10.40	0.151
Overall LOS (days)	20.32	16.11	19.54	20.97	0.065
Case dismissal					
Discharge	237	89%	278	94%	0.060
Transfer	16	6%	8	3%	
Death	14	5%	9	3%	
Morbidity	85	32%	112	38%	0.174
Non-surgical complications	52	20%	52	18%	0.552
Surgical complications	61	23%	78	27%	0.284
Bleeding	5	2%	2	1%	0.208
Sepsis	6	2%	7	2%	0.904
Aseptic wound healing disorder	6	2%	9	3%	0.539
Wound infection	8	3%	10	3%	0.771
Abdominal wall dehiscence	5	2%	4	1%	0.639
Ileus	5	2%	2	1%	0.208
Atonia (>3 days)	5	2%	16	6%	0.025
Abscess	2	1%	4	1%	0.475
Stool fistula	4	2%	0	0%	0.036
Presacral infection	4	2%	9	3%	0.213
Peritonitis	3	1%	1	0%	0.275
Colostomy complication	1	0%	4	1%	0.211
Multiple organ failure	2	1%	3	1%	0.725
Anastomotic leakage	21	12%	21	11%	0.940

**Table 4 cancers-16-01496-t004:** Presentation of histopathological findings for collectives five years before and five years after certification.

	CrCC Certification				
	−5y		+5y		*p*
Parameter	N (Mean)	% (SD)	N (Mean)	% (SD)	
Number of patients	267	47.42%	296	52.57%	
Histological verification before treatment	224	84%	272	93%	0.001
pT					
pT0	4	2%	7	3%	0.490
pT1	36	15%	28	10%	
pT2	64	26%	83	30%	
pT3	117	48%	133	48%	
pT4	22	9%	26	9%	
pN					
pN0	124	48%	170	59%	0.001
pN1	45	17%	57	20%	
pN2	58	22%	50	17%	
pNX	21	8%	5	2%	
Missing	12	5%	7	2%	
L					
L0	120	46%	168	58%	<0.001
L+	83	32%	105	36%	
Not examined	56	22%	19	7%	
V					
V0	148	57%	207	71%	
V+	47	18%	63	22%	<0.001
Not examined	63	24%	22	8%	
UICC					
I	4	2%	7	2%	<0.001
II	75	29%	82	28%	
III	43	16%	65	22%	
IV	67	25%	62	21%	
Missing	53	20%	58	20%	
M.E.R.C.U.R.Y.					
I	/	/	251	94%	/
II	/	/	10	4%	/
III	/	/	5	2%	/
Coning	/	/	6	2%	/

## Data Availability

All of the relevant data are provided in the manuscript.

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
