# Peer review of "Influence of Certification Program on Treatment Quality and Survival for Rectal Cancer Patients in Germany: Results of 13 Certified Centers in Collaboration with AN Institute"

_cancers, 2024, doi:10.3390/cancers16081496_

Round 1

Reviewer 1 Report (Previous Reviewer 1)

Comments and Suggestions for Authors

Better.

Author Response

Dear Members of Editorial Board of Cancers,

Thank you for your emails dated 8th March 2024 and 15th March 2024 offering us again the chance to revise our article Manuscript ID: cancers-2748827 entitled: ‘‘Influence of certification program on treatment quality and survival for rectal cancer in Germany. Results of 13 certified centers collaborated to AN-institute

We admire the extraordinary attention, the valuable comments and the helpful suggestions of the editors and the reviewers and we very much thank them for their important points on our manuscript.

We have revised the manuscript in light of the concerns raised, and our responses to the individual comments are detailed below:

Reviewer #1:

“Better”

Author response:

We thank you very much for your comments helping us to improve our manuscript. We hope to have met your expectations with the current revised version through implementation of all suggestions and comments of all involved reviewers. Thank you again for your contribution.

We thank you once again for consideration of our manuscript in light of our response to the reviewer`s comments.  We very much hope that you will accept the revised manuscript for publication, as we believe we have performed all proposed revisions. All revisions and corrections in the manuscript are red-highlighted.

Sincerely yours,

Aristotelis Perrakis, Mihailo Andric on behalf of the co-authors.

Reviewer 2 Report (New Reviewer)

Comments and Suggestions for Authors

This is an exciting study about a topic that would interest the readers of the journals, particularly colorectal surgeons. The paper has scientific sound; the methods are correctly used, and the results sustain the conclusions. However, a few issues should be addressed before considering the acceptance of the paper:

There is duplicate information in lines 96 - 100. There is also some redundant information regarding comparative analyses in lines 112 – 115 and 127 – 129. Please consider eliminating the redundant information.

Figure 1 is unclear in its upper part.

The studied period is 2002 to 2016. However, other information in the text suggests only ten years. Please clarify.

Please consider changing the “Kaplan-Meier diagram” to “Kaplan-Meier curves” in the statistical analysis and after that in the text.

Why do the authors use mean instead of median for the continuous variables?

The text and Table 1 of Results, Patients characteristics contain redundant information. Please consider eliminating it.

If there were no significant differences in the median number of cases between the compared groups, how does certification work for the caseload?

The text and Table 2 of Results, Perioperative parameters contain redundant information. Please consider eliminating it.

How do the authors explain the decrease of the transanal approach after certification? Is it about TaTME?

How do the authors explain the same conversion rates in laparoscopic surgery despite presumed increased experience after certification? Any data about the robotic approach? How about the increased operative time after certification?

The text and Table 3 of Results, Postoperative parameters contain redundant information. Please consider eliminating it.

How do the authors explain the relatively high postoperative LOS even after certification, with no difference from the period before certification despite the increased use of a minimally invasive approach? The same situation applies to postoperative atonia (a laparoscopic approach would decrease postoperative atonia in theory).

In lines 192 – 197, increasing/decreasing referring to the two groups of patients cannot be said if the p-value was not statistically significant.

The text and Table 4 of Results, Histopathology contain redundant information. Please consider eliminating it.

How do the authors explain better survival rates after certification?

Comments on the Quality of English Language

Minor editing of English language required

Author Response

Dear Members of Editorial Board of Cancers,

Thank you for your emails dated 8th March 2024 and 15th March 2024 offering us again the chance to revise our article Manuscript ID: cancers-2748827 entitled: ‘‘Influence of certification program on treatment quality and survival for rectal cancer in Germany. Results of 13 certified centers collaborated to AN-institute

We admire the extraordinary attention, the valuable comments and the helpful suggestions of the editors and the reviewers and we very much thank them for their important points on our manuscript.

We have revised the manuscript in light of the concerns raised, and our responses to the individual comments are detailed below:

Reviewer #2:

Author response:

Thank you very much for your valuable comments and for the time invested in detailed review. We have adjusted our paper considering your suggestions. In following, we present point-to-point reply to your concerns raised. Our changes in the manuscript are “red-highlighted”.

  1. “There is duplicate information in lines 96 - 100.

Author response:

Thank you very much for your comments and suggestions. We have followed your advice and revised our manuscript, by eliminating parts, containing duplicated information.

  1. “There is also some redundant information regarding comparative analyses in lines 112 – 115 and 127 – 129. Please consider eliminating the redundant information.”

Author response:

According to your appreciated opinion, we have eliminated the information about statistical tests performed from the lines 127-129, declared as redundant. The information in the paragraph “Statistical analysis” has also been reduced, according to your suggestion.

3.“Figure 1 is unclear in its upper part.”

Author response:

Thank you very much for your comment. We have adjusted the figure for better visualization of the investigated time frames. We hope to have brought more clearness in figure 1 in our revised version of the manuscript.

  1. “The studied period is 2002 to 2016. However, other information in the text suggests only ten years. Please clarify.”

Author response:

Thank you for your question, please allow us to clarify.

As displayed in lines 104-108, we raised data of surgically treated patients for rectal cancer during the five-year period before the certification date (included to -5y group) and for five-year period after the certification date (included to +5y group), for every single included center. For example, if an oncological unit became certified Colorectal Cancer Center in 2011, the included cases from this particular center, came from the period 2006-2011 (included to -5y group) and 2011-2016 (included to +5y group). Equivalently, if a certification took place in 2007, the included cases came from periods 2002-2007 (-5y) and 2007-2012 (+5y).

In overall, we included patients which were surgically treated in the period 2002-2016 (as displayed in the line 92-93), meaning that the certification of all included centers took place between the years 2007 and 2011.

For better understanding, we brought some changes in the paper on this topic, which are red-highlighted.

5.“Please consider changing the “Kaplan-Meier diagram” to “Kaplan-Meier curves” in the statistical analysis and after that in the text.”

Author response:

Thank you for your valuable comment. We fully agree with you and we have performed changes to Kaplan-Meier curves, as suggested.

6.“Why do the authors use mean instead of median for the continuous variables?”

Author response:

Thank you very much for addressing this point. For the presentation of the continuous variables, we used the Mean and SD, as the data were symmetrical, which is a valid way of presentation in that case. Otherwise, the Median and Range would be an adequate alternative, as you suggested. The case has been validated by our statisticians. Thank you very much for your understanding.

7.“The text and Table 1 of Results, Patients characteristics contain redundant information. Please consider eliminating it.”

Author response:

Thank you very much for your valuable comment. In order to remove redundant information, the text in this paragraph has been relevantly trimmed and marked red. For better presentation, the order of included data has been adjusted and demonstrated in the table 1.

8.“If there were no significant differences in the median number of cases between the compared groups, how does certification work for the caseload?”

Author response:

Thank you for your valuable comment.

Although the centralization and increasing the caseload in certified centers is one of wished effects of the certification, this aspect obviously has not been observed in the investigated centers during the study period. For your understanding: every hospital in Germany is allowed to treat patients with colorectal cancer regardless of certification as CrCC. In other words, colorectal centers in Germany are not obligated to be certified. Therefore, more than half of the patients are still being treated in non-certified centers (53% in 2017, probably much more during the presented study period). Latter phenomenon affects the case load in some centers, despite being certified as CrCC. In the last 3 years there is a great debate about this aspect in Germany and the main intention of the German Cancer Society and of further institutions is to set stricter quality criteria for hospitals and/or centers treating patients for colorectal cancers, mainly in terms of certification. This discussion about quality of treatment and the valuable aspect of certification has been the driving force for us to perform this study and to show the advantages of certification. Furthermore, we unfortunately do not have data available now about the further development of the caseload in the investigated centers after 2016. We have already requested this data and our intention is to further publish about this very important subject. Thank you again for your valuable comments and questions.

9.“The text and Table 2 of Results, Perioperative parameters contain redundant information. Please consider eliminating it.”

Author response:

Thank you very much for the valuable hint. We have removed redundant information from the text in this paragraph, as well as from the table 2.

10.How do the authors explain the decrease of the transanal approach after certification? Is it about TaTME?

Author response:

Thank you very much for this question. We have already discussed this topic in the lines 360-366 considering local excisions of rectal cancer and revised the discussion section, in order to answer your question and make this aspect clear:

… We furthermore report a significant reduction in transanal excisions (8% to 1%, p<0.001) performed after CrCC-certification. As the S3-guideline clearly recommends neoadjuvant chemoradiotherapy followed bei total mesoractal excision (TME) for most of the cases of low rectal cancers [6] and we witnessed an expansive development of endoscopic resections for adenoma and early rectal cancers in last years [30], we assume that over the timeline more patients were indicated one of the above mentioned treatment options, which resulted in decrease of local surgical excisions….

We unfortunately did not have any data about TaTME in the examined centers.

11.“How do the authors explain the same conversion rates in laparoscopic surgery despite presumed increased experience after certification? Any data about the robotic approach? How about the increased operative time after certification?”

Author response:

We thank you for addressing this topic. We revised the discussion section according to your comments and further discussed this matter, which was mostly addressed in the discussion (lines 346-359) and lines 367-369:

  1. “Our analysis showed an increasing proportion of laparoscopic approach from 5% to 55% during five-year period after certification (p<0.001), without increase in conversion rate (p=0.504). Although the certification program does not directly obligate certified centers to perform a certain proportion of minimally-invasive approach for rectal surgery, ……as the robotic colorectal surgery developed in the included centers after the investigated period…”
  2. “The significant prolongation of operating time after certification could be explained due to increase of laparoscopic approach. Prolonged operating times in laparoscopic cases are well known and already reported in several studies [26]

Regarding stable conversion rate despite increasing numbers in laparoscopic approach, we do not have any applicable data, to certainly clarify this dilemma. We have indeed observed the increase of laparoscopic approach, but we must concern the learning curve, which possibly affects the data.

On the other hand, according to our experience, with increasing laparoscopic experience, also the complexity of the laparoscopically approached cases increases, so this could possibly be another relevant aspect. Robotic colorectal surgery has not been in focus in Germany, considering the study time period. The current trend, concerning robotic colorectal procedures started in Germany mainly since 2017.

12.“The text and Table 3 of Results, Postoperative parameters contain redundant information. Please consider eliminating it.”

Author response:

Thank you for this comment. Following your suggestion, we have reduced the presented information in the text and table 3. However, several variables described in this paragraph have been processed in the discussion, so we found it relevant to keep some of the information in the table 3. We hope for your understanding.

13.“How do the authors explain the relatively high postoperative LOS even after certification, with no difference from the period before certification despite the increased use of a minimally invasive approach? The same situation applies to postoperative atonia (a laparoscopic approach would decrease postoperative atonia in theory).”

Author response:

We thank you for pointing to this important observation. We also have found it relevant and therefore have discussed it in the revised version of our manuscript in the lines 376-390 and lines 407-413:

  1. “ As a further aspect of improved management triggered by the certification process, there have been several organizational advantages observed during the period after certification: the preoperative length of in-hospital stay has been significantly reduced after certification. Furthermore, we observed the trend to reduction of postoperative length of in-hospital stay. Additionally. there was a clear tendency of shortening overall length of in-hospital stay during the period after certification, comparable to other studies [35]…….. Although here shown time frames represent the historic philosophy of perioperative management (diagnostics under stationary conditions, long in-house stays until wound sutures are removed), which strongly varies from nowadays strived fast-track surgery [36-38], the data show obvious development after certification in terms of shortened in-hospital stays.”
  2. “On the other side, during the five-year period after CrCC-certification there has been a significant increase of postoperative intestinal atonia. This observation is contradictory to results of different studies, who suggest faster postoperative bowel movement after laparoscopic approach [39, 40]. This result could eventually be the result of the standard postoperative analgetic regimen, which was mainly opioid-based in the most centers, associated with AN-Institute, included to our collective, which lead to increased intestinal atonia, as reported in several studies [41]

  1. In lines 192 – 197, increasing/decreasing referring to the two groups of patients cannot be said if the p-value was not statistically significant.

Author response:

We thank you for this hint. Considering your suggestion, we adjusted the formulation in this paragraph.

15.“The text and Table 4 of Results, Histopathology contain redundant information. Please consider eliminating it.”

Author response:

Thank you for this valuable comment.

We thank you for your valuable comments and respect your opinion on this topic. We have reduced the presented data in text and in the table 3. On the other hand, the development in histopathological documentation quality is one of the important aspects discussed as an effect of certification process. Therefore, a relevant part of the histopathological data had to be contained in this paragraph. We hope for your understanding.

  1. How do the authors explain better survival rates after certification?”

Author response:

We thank for this important question. We added a comment to the improved survival in the lines 453-456.

We interpreted this observation as a result of multifactorial/multidisciplinary process including increasing quality in diagnostics, choosing adequate treatment modality and improved surgical technique, multidisciplinary treatment modalities, increased quality of histopathological documentation, as well as further not measurable aspects, all indicated by the certification process. For better understanding, we added a paragraph to the dicussion chapter in the lines 468-473.

In summary, we admire your comments about “redundant information”, which were possibly overloading our paper. Therefore, we significantly trimmed the manuscript fully under consideration of your points during all review rounds. We focused on the quantitative and qualitative aspects involved within the scope of the structured certification program in Germany and underlined the most important objectives, which are now clearly mentioned in the current version of the manuscript. All changes in data distribution, data analysis, results, discussion, and conclusions are “red highlighted”.

We thank you once again for consideration of our manuscript in light of our response to the reviewer`s comments.  We very much hope that you will accept the revised manuscript for publication, as we believe we have performed all proposed revisions. All revisions and corrections in the manuscript are red-highlighted.

Sincerely yours,

Aristotelis Perrakis, Mihailo Andric on behalf of the co-authors.

Round 2

Reviewer 2 Report (New Reviewer)

Comments and Suggestions for Authors

The authors correctly addressed all major concerns raised by the reviewers

Comments on the Quality of English Language

Minor editing of English language required

This manuscript is a resubmission of an earlier submission. The following is a list of the peer review reports and author responses from that submission.

Round 1

Reviewer 1 Report

Comments and Suggestions for Authors

I don't understand the difference between the A and B group hospitals. All of them were certified. Which criteria was used to assign a hospital to group A or B.

I suppose the analysis could be done for all results between patients treated 5 or 3 years before and after certification. It means, you could perform analysis of all 1188 patients divided in 2 groups with better results.

Author Response

Reviewer #1:

„I don't understand the difference between the A and B group hospitals. All of them were certified. Which criteria was used to assign a hospital to group A or B..”

Author response:

We thank you very much for your comment and appreciate your concern. Group A and Group B concern the same hospitals. The only difference between the two groups has to do with the period before and after certification:

  1. Group A: 3 years before and 3 years after certification
  2. Group B: 5 years before and 5 years after certification. We clarify this matter by adding a sentence in the manuscript body and Figure 1 (page 3), presenting the stratification procedure.

„I suppose the analysis could be done for all results between patients treated 5 or 3 years before and after certification. It means, you could perform analysis of all 1188 patients divided in 2 groups with better results.“

Author response:

We thank you very much for your comment and ant to clarify your concern concerning the both collectives. Overall, we analyzed 625 patients, receiving treatment from 2002 to 2016. In order to prevent any kind of bias, considering the heterogeneity of data over the years, especially before the implementation of certification processes from the German Cancer Society and to have a clear trend in our comparative analysis in terms of short-(perioperative), middle- and long-term-setting (5-years follow-up after treatment), we stratified patients in two groups:

  1. Those who underwent treatment for rectal cancer three years before and three years after certification (Group A, n=625).
  2. Those patient underwent treatment five years before and five years after certification (Group B, n= 563) . To clarify this matter and to make our stratification process clear, we added a paragraph in the manuscript main body.

Reviewer 2 Report

Comments and Suggestions for Authors

Dear Author

I read your article with great pleasure, in particular because of the importance of improving quality in centers that perform colorectal surgery.

The question arises spontaneously whether, before being accredited, the centers analyzed were already considered colorectal centres; this is because it could explain the lack of statistically significant values between before and after certification.

CORRECTIONS TO THE ARTICLE:

pag3 ......B-3 and B+3 need correction B+5-5

pag7...... L-staus and V-status....... need to be specified at this point (see pag 17)

pag16....... course including complications, inovolving physiotherapy (involving)......................... increase off postoperative (off.... of)

pag17...., die MERCURY I rate amounted 96.4% (die.... the)

FIGURE 2-3-4-5 need to be translated into English

QUESTIONS TO THE ARTICLE

pag4 Perioperative parameters..... has the number of patients who underwent preoperative and postoperative neoadjuvant treatment changed?

pag4.... transanal approach for excision of rectum carcinoma significantly decreased after the certification..... How can this reduction be explained?

discontinuous resections according to Hartmann increased significantly...... has it been verified whether there is a correlation with ASA score, age, comorbidity, transanal approach?

pag9....... In further subgroup analysis, significant improvement of overall survival could...... Which is this further subgroup?

pag10.... Survival according to UICC stage before and after certification (collective A)

In the comparison of particular UICC stages between the three year period before certification (A-3) and three year period after certification (A+3), the analysis showed nonsignificantly better overall survival for UICC stages I, III and IV (A: I : p=0.147, III: p=0.494, IV: p=0.759) and similar overall survival for UICC stage II (II: p=0.349), (Figure 8).

Regarding disease free survival the analysis showed a significant improvement for UICC stage I within collective A due to certification (A I: p=0.040), but no significant difference for UICC stages II and III (A: II: p=0.483, III: p =0.567)........ What is the interpretation of this data? If in the earliest stage (STAGE1) the number of lymph nodes analyzed or other factors have changed...

pag15... significant increase of discontinuous resections according to Hartmann ..... Have you checked whether there has been an increase in reversal of Hartmann procedures?

of the fear of anastomotic insufficiency....... is the meaning "insufficiency" as leakage? "fear" does not seem adequate, but is necessary if one of the factors for accreditation is the reduction of the rate of anastomotic fistulas. In this case it must be clarified

pag17.... MERCURY need a citation in references

About Limitation:

  still missing relevant data for treatment of rectal cancer, such as neoadjuvant and adjuvant treatment.

Have you checked whether this data is traceable after the moment of certification?

Comments for the author

Although the study is very interesting and a reason for improvement, I consider the missing data on adjuvant and neoadjuvant treatment to be an important limiting factor. I believe the verification of this data after the moment of accreditation is fundamental.

Comments on the Quality of English Language

The quality of English is good, and minor revision is required. It is necessary to translate some figures in English.

Author Response

Reviewer #2:

„I read your article with great pleasure, in particular because of the importance of improving quality in centers that perform colorectal surgery. The question arises spontaneously whether, before being accredited, the centers analyzed were already considered colorectal centres; this is because it could explain the lack of statistically significant values between before and after certification.“

Author response:

We thank you very much for your comments and your remark. We agree with your concern. However, the treatment of colorectal cancer in Germany does not take place only in colorectal centers. That means a patient with colorectal cancer may receive treatment in a hospital  irrespective of medical care level (basic, priority and maximal care hospitals) and of presence of colorectal center or not. Therefore, we performed this analysis concerning medical care level and certification status.

„pag7...... L-staus and V-status....... need to be specified at this point (see pag 17)“

Author response:

We thank you very much for this point. We agree with the concern raised from your side. Therefore, we revised this matter by adding a paragraph concerning L-status and V-status on page 7.

 Perioperative parameters..... has the number of patients who underwent preoperative and postoperative neoadjuvant treatment changed?“

Author response:

We thank you very much for your comment and appreciate your concern. Unfortunately, we cannot answer your question, concerning the available data. Regarding preoperative (neoadjuvant) and postoperative (adjuvant) treatment we did not have enough data concerning the patients received treatment before certification. This data was available for our collectives 3 years and 5 years after certification, respectively. This matter is in our honest opinion the most important limitation factor in our study. This matter concerns your further questions and your final remark. In order to clarify this situation, we added a sentence in the “Limitation-Section” pointing out the importance of this matter and a paragraph in the “Results Section”.

„pag4.... transanal approach for excision of rectum carcinoma significantly decreased after the certification..... How can this reduction be explained?“

 Author response:

We thank you very much for this point. We agree with the concern raised from your side. However, we believe that this matter is completely incidental and we have no valid explanation for this decrease of transanal excisions.

Point 9: „discontinuous resections according to Hartmann increased significantly...... has it been verified whether there is a correlation with ASA score, age, comorbidity, transanal approach?“

 Author response:

We thank you very much for this point. After thorough research and the performed statistical analysis, we did not find any significant correlation with ASA score, age, comorbidity and transanal approach. However, we observed a trend (non-significant) for more emergency cases. In order to clarify this point, we added a sentence in the “Discussion Section” (“red-highlighted”)

„pag9....... In further subgroup analysis, significant improvement of overall survival could...... Which is this further subgroup?“

Author response:

We thank you very much for this point. The performed subgroup analysis concerns the medical care level (basic, priority and maximal care hospitals) of the 13 examined hospital. At the mentioned point, we concern the subgroup analysis of the maximal medical care hospitals. Therefore, we added a sentence in order to make this point clear.

„pag10.... Survival according to UICC stage before and after certification (collective A) In the comparison of particular UICC stages between the three year period before certification (A-3) and three year period after certification (A+3), the analysis showed nonsignificantly better overall survival for UICC stages I, III and IV (A: I : p=0.147, III: p=0.494, IV: p=0.759) and similar overall survival for UICC stage II (II: p=0.349), (Figure 8).

 Regarding disease free survival the analysis showed a significant improvement for UICC stage I within collective A due to certification (A I: p=0.040), but no significant difference for UICC stages II and III (A: II: p=0.483, III: p =0.567)........ What is the interpretation of this data? If in the earliest stage (STAGE1) the number of lymph nodes analyzed or other factors have changed...“

 Author response:

We thank you very much for this point. We agree with the concern raised from your side. However, in the performed analysis we could not record any significant changes as far as lymph node harvesting or other histopathological factors ae concerned. Nevertheless, we did not have the M.E.R.C.U.R.Y. classification of the specimens before certification and we believe that this matter may concern the surgical quality performed for small rectal cancers. Unfortunately, we cannot postulate such a matter after not having this important data available.

„pag15... significant increase of discontinuous resections according to Hartmann ..... Have you checked whether there has been an increase in reversal of Hartmann procedures?

Author response:

We thank you very much for this point. After thorough research, we recorded a “logical” increase of reversal of Hartmann. Overall 75% of the patients received a discontinuous resection, also received a reversal of Hartmann, 6-12 months after initial treatment.

„of the fear of anastomotic insufficiency....... is the meaning "insufficiency" as leakage? "fear" does not seem adequate, but is necessary if one of the factors for accreditation is the reduction of the rate of anastomotic fistulas. In this case it must be clarified“

Author response:

We thank you very much for this point. We agree with the concern raised from your side. Firstly, we performed a thorough spelling check and we replaced the term: “insufficiency” with the term “leakage”. According to the Certification Requirements set by the German Cancer Society, the maximal rate of anastomotic leakage allowed is 15% of all patients underwent surgery. Therefore, we firmly believe that this aspect (“fear of anastomotic leakage”) is being taking under consideration, before performing an anastomosis in a high-risk situation (such as emergency). Latter has already been reported in other studies, which have been cited.

About Limitation: still missing relevant data for treatment of rectal cancer, such as neoadjuvant and adjuvant treatment. Have you checked whether this data is traceable after the moment of certification?

 Author response:

We thank you very much for this point. We understand this major concern raised from your side. Overall we can report that all patients with an advanced rectal carcinoma (cT3/cT4 and/or CN+) , treated in the time frame after certification received a neoadjuvant chemoradiation and patients with nodal-positive status in the histopathological examination an adjuvant chemotherapy as defined by the S3-german guidelines. As far as the patients treated before certification are concerned, we do not have enough data neither for neoadjuvant nor for adjuvant treatment and therefore we cannot perform a comparative analysis. This aspect has been clarified in the revised manuscript.

pag3 ......B-3 and B+3 need correction B+5-5“

„pag16....... course including complications, inovolving physiotherapy (involving)......................... increase off postoperative (off.... of)“

„ pag17...., die MERCURY I rate amounted 96.4% (die.... the)“

„FIGURE 2-3-4-5 need to be translated into English“

„pag17.... MERCURY need a citation in references“

“The quality of English is good, and minor revision is required. It is necessary to translate some figures in English.“

Author response:

We thank you for these points and appreciate your valuable comments. We agree with your concerns and performed a thorough spelling check. All several minor edition corrections  (citation of M.E.R.C.U.R.Y. and translation of figures) have been performed as suggested.

„Although the study is very interesting and a reason for improvement, I consider the missing data on adjuvant and neoadjuvant treatment to be an important limiting factor. I believe the verification of this data after the moment of accreditation is fundamental.

 Author response:

We thank you very much for this point. We understand this major concern raised from your side. Therefore, we attenuate our conclusions by adding a new sentence and underlining again the limitations of our study. A complete revision of the “conclusion-section” in this context has been also performed.

We thank you very much for your comments in the peer-review process, helping us making this manuscript better.

Reviewer 3 Report

Comments and Suggestions for Authors

1 The research method, the author uses grouping A and B, is really difficult to understand. Can you explain the reason in detail?

2 This study is actually a data mining study of a multi-center database. The explanation of the basic quality of the data needs to be explained, or whether corresponding reports can be provided to explain it, so as to facilitate understanding of the basic status of the multi-center database.

3 In the results section, it is mentioned: “Regarding the appearance of intraoperative complications (tumor perforation, bleeding, organ lesions), there was a significant decrease in the collective A after certification, while there was no significant change within the collective B”. What does "organ lesions" mean here? In addition, “intraoperative complications” generally refer to intraoperative complications, which specifically refers to problems related to surgery. What is mentioned in the paper are actually complications associated with tumors.

4 Is the abbreviation “praop.LOS” mentioned in the result 3.3 wrong?

5 At the same time, what specific situation does “stool fistula” in 3.3 refer to? Does it involve anastomotic leakage? If it is really anastomotic leakage, there will be no anastomotic leakage in the control group after surgery. This is a very special situation and the data needs to be verified.

6 In 3.4 "The distribution of patients regarding pathological T-stadium did not change within collectives A and B after the certification for CrCC". Is there an error in the T staging? The following paragraphs also have similar situations.

7 In Table 4, the author mentions Histologicaly verified carcinoma, this expression is slightly awkward. Is it possible to adjust, and in addition, since the data sources in the study are all patients with rectal cancer, why is it necessary to provide such information here?

8 In the same table, MERCURY mentions “Coning”, what does it mean here?

9 During the discussion, the content discussed by the author was quite confusing. I felt that a lot of the results of this data analysis were listed. It is recommended to make appropriate summary and form some valuable discussion points for the conclusion, so that readers can better understand.

Author Response

Reviewer #3:

  1. “The research method, the author uses grouping A and B, is really difficult to understand. Can you explain the reason in detail?”

 Author response:

We thank you very much for your comment and ant to clarify your concern concerning the both collectives. Overall, we analyzed 625 patients, receiving treatment from 2002 to 2016. In order to prevent any kind of bias, considering the heterogeneity of data over the years, especially before the implementation of certification processes from the German Cancer Society and to have a clear trend in our comparative analysis in terms of short-(perioperative), middle- and long-term-setting (5-years follow-up after treatment), we stratified patients in two groups:

  1. Those who underwent treatment for rectal cancer three years before and three years after certification (Group A, n=625).
  2. Those patient underwent treatment five years before and five years after certification (Group B, n= 563) . To clarify this matter and to make our stratification process clear, we added a paragraph in the manuscript main body.

  1. “This study is actually a data mining study of a multi-center database. The explanation of the basic quality of the data needs to be explained, or whether corresponding reports can be provided to explain it, so as to facilitate understanding of the basic status of the multi-center database.”

Author response:

We thank you very much for this point. We can understand the concern raised from your side. This study is -as described- a comparative study in retrospective fashion, from a prospectively acquired, multicentre database from the AN-Institute of Otto-von-Guericke University in Magdeburg, Germany. All data has been acquired from the 13 associated hospitals, based on standardized documentation forms, which have been drafted by the scientific advisory board of AN-institute. Of course, there is a heterogeneity as far as the data collection and quality of data are concerned. Main reason for this is the dynamic change of gold standards in treatment of rectal cancer (see neoadjuvant chemoradiation, the meaning of M.E.R.C.U.R.Y. classification for surgical quality and its impact of reccurence rates etc.). The breakthrough in terms of quality in data sampling and tumor documentation in Germany has been the implementation of certification process for Colorectal Centers by the German Cancer Society. Latter has been the driver for the Centers to optimize the databases and the parameters raised. Since 2010 the scientific advisory board of AN-institute revised all documentation forms for tumor entities according to the high standards, defined by the German Cancer Society. Overall, we performed this analysis, pointing out this heterogeneity as a limitation of our study and also underlining the importance of certification process also in terms of significant better and more scientific tumor documentation. Thank you for this important point. We added a paragraph in the Sections: “Material and Methods” and “Limitation” in order to clarify this important point  

  1. „In the results section, it is mentioned: “Regarding the appearance of intraoperative complications (tumor perforation, bleeding, organ lesions), there was a significant decrease in the collective A after certification, while there was no significant change within the collective B”. What does "organ lesions" mean here? In addition, “intraoperative complications” generally refer to intraoperative complications, which specifically refers to problems related to surgery. What is mentioned in the paper are actually complications associated with tumors.“

 Author response:

We thank you very much for this point. We agree with the concern raised from your side. We re-write this part by replacing the term: “organ lesions” with the term “injury of adjacent organs and/or main structures. Furthermore, we rephrase this paragraph in order to clarify what was meant by “intraoperative complications”.

  1. „Is the abbreviation “praop.LOS” mentioned in the result 3.3 wrong?“

 Author response:

We thank you for this point and appreciate your valuable comments. We agree with your concern and performed a thorough spelling check. All several minor edition corrections have been performed as suggested.

  1. “At the same time, what specific situation does “stool fistula” in 3.3 refer to? Does it involve anastomotic leakage? If it is really anastomotic leakage, there will be no anastomotic leakage in the control group after surgery. This is a very special situation and the data needs to be verified.”

 Author response:

We thank you very much for this point. We understand the concern raised from your side. “Stool fistula” was defined as a fistula coming from all other parts of bowel (small and large bowel), but not from the anastomosis. Both anastomotic leakage and/or fistula coming from anastomosis, in either short-, or long-term setting have been classified as anastomotic leakage. This point has been clarified in the revised version of the manuscript.

  1. „In 3.4 "The distribution of patients regarding pathological T-stadium did not change within collectives A and B after the certification for CrCC". Is there an error in the T staging? The following paragraphs also have similar situations.“

Author response:

We thank you for this point and appreciate your concern and your point. We firmly believe that there is a misunderstanding because of a spelling mistake from our side. What was meant was that in terms of pT- stages there was a homogeneity of the collectives. As far as the pN-status was concerned, there was a significant increase of pN0 and pN1 cases with synchronous significant decrease of pN2-cases. Regarding patients with distant metastasis  there was a trend for decrease within Group A and a significant decrease within Group B. We agree with your concern. Therefore, we performed a thorough spelling check and rephrased the whole paragraph.

  1. „In Table 4, the author mentions Histologicaly verified carcinoma, this expression is slightly awkward. Is it possible to adjust, and in addition, since the data sources in the study are all patients with rectal cancer, why is it necessary to provide such information here?“

We thank you for this point and appreciate your valuable comment. We rephrased this (“histological verification before treatment”, see revised manuscript), as suggersted. We firmly believe that this information is necessary and important, because in the era before certification process, the histological verification before treatment has not been standardized. The histopathological verification before start of the treatment is an important parameter in the certification process and we wanted to examine if there was a trend towards this direction in our analysis.

  1. „In the same table, MERCURY mentions “Coning”, what does it mean here?“

Author response:

We thank you very much for this point. Coning refers to the tendency for the surgeon to cut towards the tubular rectum during distal dissection, rather than staying outside the visceral mesorectal fascia; coning gives the specimen a tapered, conical appearance and is an indication of suboptimal surgical quality, as described by Hermanek et al. (Hermanek P, Hermanek P, Klimpfinger M, et al. The pathological assessment of mesorectal excision: implications for further treatment and quality management. In J Colorectal Dis 2003;18:335–41.) This is a further established histopatholgical aspect indicating quality of surgery. Latter is together with M.E.R.C.U.R.Y. the most important criteria for surgical quality in rectal surgery, as defined by the German Cancer Society and therefore an indispensable surgical aspect in the certification process for CrCC. We added a citation in our references in order to clarify this aspect

  1. „During the discussion, the content discussed by the author was quite confusing. I felt that a lot of the results of this data analysis were listed. It is recommended to make appropriate summary and form some valuable discussion points for the conclusion, so that readers can better understand.“

 Author response:

We thank you very much for this point. We understand this major concern raised from your side and therefore we revised the Section: “Discussion”.

We thank you for the valuable comments, we accept the concerns of the reviewers, we revised the manuscript, and all corrections have been red highlighted. We also performed all suggested revisions from MDPI editorial office (simple summary added, use of the suggested reference format, revision of IRB- and IC-statement)

We thank you once again for consideration of our manuscript in light of our response to the reviewer`s comments. 

We very much hope that you will accept the revised manuscript for publication, as we believe we have performed all proposed revisions. All revisions and corrections in the manuscript are red highlighted.

Sincerely yours,

Aristotelis Perrakis, Mihailo Andric on behalf of the co-authors.

Round 2

Reviewer 1 Report

Comments and Suggestions for Authors

I didn't get the clear answer why authors compare 4 similar groups and distribute the numbers instead of comparing the 2 groups.

The separation is the same - before and after certification. Time if follow-up, according to my mind, doesn't matter because statistical programs can easily to manage this problem and give better answers.

I repeat my unchanged opinion - this work is too complicated and unclear.

I doesn't understand the main idea. Idea that sertification is good can be proved by analasing the B group without the bringing confusion in A group data analysis.

It is obvios that during longer follow up you will get more statistical significant results. And we see it in your calculations.

Your work can be published only after redoing it and leaving only two groups to compare. Because now it is great mess.

Author Response

Dear Members of Editorial Board of Cancers,

Thank you for your email dated the 13th  December 2023 offering us again the chance to revise and re-submit our article Manuscript ID: cancers-2748827 entitled: ‘‘Influence of certification programme on treatment quality and survival for rectal cancer in Germany. Results of 13 cerftified centers collaborated to AN-institute

We admire the extraordinary attention, the valuable comments and the helpful suggestions of the editors and the reviewers and we very much thank them for their important points on our manuscript.

We have revised the manuscript in light of the concerns raised, and our responses to the individual comments are detailed below:

Reviewer #1:

„I didn't get the clear answer why authors compare 4 similar groups and distribute the numbers instead of comparing the 2 groups.

The separation is the same - before and after certification. Time if follow-up, according to my mind, doesn't matter because statistical programs can easily to manage this problem and give better answers.

I repeat my unchanged opinion - this work is too complicated and unclear.

I doesn't understand the main idea. Idea that sertification is good can be proved by analasing the B group without the bringing confusion in A group data analysis.

It is obvios that during longer follow up you will get more statistical significant results. And we see it in your calculations.

Your work can be published only after redoing it and leaving only two groups to compare. Because now it is great mess..”

Author response:

We thank you very much for your comments, admire your great interest and appreciate your opinion and concerns.

Firstly, we have to make clear that the questions raised in the current (second) review round are new. The raised questions in the first review round have been answered clearly, in our honest opinion. The questions raised from your side in the first review concerned the distribution of both collectives and how we stratified groups A and B and which patients have been analyzed in the two groups:

(Original answer from first revision: “We thank you very much for your comment and appreciate your concern. Group A and Group B concern the same hospitals. The only difference between the two groups has to do with the period before and after certification:

  1. Group A: 3 years before and 3 years after certification
  2. Group B: 5 years before and 5 years after certification. We clarify this matter by adding a sentence in the manuscript body and Figure 1 (page 3), presenting the stratification procedure.”).

In the current review round and without any comment in the first review, you consider the whole idea of this manuscript as not acceptable and ask us for re-doing the whole analysis. We admire your great interest in making our paper better. Nevertheless, we have to underline that the main idea behind this distribution is to present the advantages, offered by a certification process, in a dynamic fashion, considering a timeline. Therefore, we offer the differences seen in overall 13 centers in a timeline, considering the most important milestones namely 3 and 5 years. As you certainly know, as a famous colorectal specialist, the most critical time for the occurrence of locoregional recurrences is 3 years after primary treatment (surgery with/without neoadjuvant chemoradiation in a multimodal fashion) with 70% of occurrence within the first 3 years after treatment. And of course the further milestone for distribution: five years in order to collect valid information and to perform valid statistics concerning 5 year overall- and tumor-free survivals etc. What you propose is a static, dry setting considering only the five years before and after certification, which does not fit to the nature of this paper. We accept your comment that the current setting might be more challenging to understand, but on the other hand we consider this paper as a tool for colorectal specialists, indicating the overall advantages (qualitative and quantitative) offered by certification. In such a setting, we firmly believe that such a work has to be published in its current dynamic setup and format. In order to make you feel comfortable and your paper more attractive to read we added a paragraph, indicating why it is essential to have a dynamic setup in this paper.

 Further remarks from your side:

“The separation is the same - before and after certification. Time if follow-up, according to my mind, doesn't matter because statistical programs can easily to manage this problem and give better answers.”

This point from your side is extremely valuable because we firmly believe that only statistical programs, because of the multivariability of quantitative and qualitative aspects involved, cannot manage a dynamic evolution process within the scope of a structured certification program. Latter aspect makes a dynamic and continuous evaluation over a timeline indispensable. The 3-year and 5 year-milestones are furthermore the most important ones, set by the German Cancer Society: Within the scope of a certification program, the German Cancer Society is performing the first follow-up controls in a certified center after the first 3 years, involving all parameters analyzed in this paper. It is not a matter of “calculations” and “getting more significant statistics” as you mentioned in your valued comment. It is about the evaluation of an evolution process, considering changing of “oncological mind” and bringing quality and multimodal approaches for the best of patients suffering from colorectal cancer. Moreover, this one must be dynamic. We firmly believe that our readers, as colorectal specialists will appreciate this dynamic setting of the manuscript, as the further 2 reviewers already did.

We thank you for the valuable comments, we accept the concerns of the reviewers, we revised the manuscript, and all corrections of the second review round have been green-highlighted.

All already performed corrections for the first review-round remained red-highlighted.

We thank you once again for consideration of our manuscript in light of our response to the reviewer`s comments. 

We very much hope that you will accept the revised manuscript for publication, as we believe we have performed all proposed revisions. All revisions and corrections in the manuscript are red- (first review round) and green-(second review round)-highlighted.

Sincerely yours,

Aristotelis Perrakis, Mihailo Andric on behalf of the co-authors.

Reviewer 2 Report

Comments and Suggestions for Authors
Dear Author 
I appreciate your editing and remarking of the partial limitation of the study, concerning the preoperative treatment... but it has become an important improvement for patients.
As you answered to Hartmann procedure "Overall 75% of the patients received a discontinuous resection, also received a reversal of Hartmann, 6-12 months after initial treatment.", consider that in the literature reversal Hartmann is about 26-60%. You can speculate that a more careful and thoughtful approach was adopted  (see references above:  9. Whitney S, Gross BD, Mui A, Hahn S, Read B, Bauer J. Hartmann's reversal:Factors affecting complications and outcomes. Int J Colorectal Dis. 2020;35:1875–80. [PubMed] [Google Scholar] 10. Hallam S, Mothe BS, Tirumulaju R. Hartmann's procedure, reversal and rate of stoma-free survival. Ann R Coll Surg Engl. 2018;100:301–7. [PMC free article] [PubMed] [Google Scholar]11. Aydin HN, Remzi FH, Tekkis PP, Fazio VW. Hartmann's reversal is associated with high postoperative adverse events. Dis Colon Rectum. 2005;48:2117–26. [PubMed] [Google Scholar]12. Okolica D, Bishawi M, Karas JR, Reed JF, Hussain F, Bergamaschi R. Factors influencing postoperative adverse events after Hartmann's reversal. Colorectal Dis. 2012;14:369–73. [PubMed] [Google Scholar]13. Cellini C, Deeb AP, Sharma A, Monson JR, Fleming FJ. Association between operative approach and complications in patients undergoing Hartmann's reversal. Br J Surg. 2013;100:1094–9. [PubMed] [Google Scholar]14. Vasiliu EC, Zarnescu NO, Costea R, Rahau L, Neagu S. Morbidity after reversal of Hartmann operation:Retrospective analysis of 56 patients. J Med Life. 2015;8:488–91. [PMC free article] [PubMed] [Google Scholar]

It would be important (I apologize if I did not ask you before) to enter the median number of cases treated by the centers (colon and rectal cancers). It would be essential to understand whether certification and/or volume alone are factors for improvement.

Author Response

Dear Members of Editorial Board of Cancers,

Thank you for your email dated the 13th  December 2023 offering us again the chance to revise and re-submit our article Manuscript ID: cancers-2748827 entitled: ‘‘Influence of certification program on treatment quality and survival for rectal cancer in Germany. Results of 13 cerftified centers collaborated to AN-institute

We admire the extraordinary attention, the valuable comments and the helpful suggestions of the editors and the reviewers and we very much thank them for their important points on our manuscript.

We have revised the manuscript in light of the concerns raised, and our responses to the individual comments are detailed below:

Reviewer #2:

Dear Author 

I appreciate your editing and remarking of the partial limitation of the study, concerning the preoperative treatment... but it has become an important improvement for patients.

As you answered to Hartmann procedure "Overall 75% of the patients received a discontinuous resection, also received a reversal of Hartmann, 6-12 months after initial treatment.", consider that in the literature reversal Hartmann is about 26-60%. You can speculate that a more careful and thoughtful approach was adopted  (see references above:  9. Whitney S, Gross BD, Mui A, Hahn S, Read B, Bauer J. Hartmann's reversal:Factors affecting complications and outcomes. Int J Colorectal Dis. 2020;35:1875–80. [PubMed] [Google Scholar] 10. Hallam S, Mothe BS, Tirumulaju R. Hartmann's procedure, reversal and rate of stoma-free survival. Ann R Coll Surg Engl. 2018;100:301–7. [PMC free article] [PubMed] [Google Scholar]11. Aydin HN, Remzi FH, Tekkis PP, Fazio VW. Hartmann's reversal is associated with high postoperative adverse events. Dis Colon Rectum. 2005;48:2117–26. [PubMed] [Google Scholar]12. Okolica D, Bishawi M, Karas JR, Reed JF, Hussain F, Bergamaschi R. Factors influencing postoperative adverse events after Hartmann's reversal. Colorectal Dis. 2012;14:369–73. [PubMed] [Google Scholar]13. Cellini C, Deeb AP, Sharma A, Monson JR, Fleming FJ. Association between operative approach and complications in patients undergoing Hartmann's reversal. Br J Surg. 2013;100:1094–9. [PubMed] [Google Scholar]14. Vasiliu EC, Zarnescu NO, Costea R, Rahau L, Neagu S. Morbidity after reversal of Hartmann operation:Retrospective analysis of 56 patients. J Med Life. 2015;8:488–91. [PMC free article] [PubMed] [Google Scholar]

It would be important (I apologize if I did not ask you before) to enter the median number of cases treated by the centers (colon and rectal cancers). It would be essential to understand whether certification and/or volume alone are factors for improvement.

Author response:

We thank you very much for your comments and your appreciation towards our efforts, making this paper more interesting for our valuable readers. Considering your comment concerning the very high rate of Hartmann-reversals in our study, we agree with your comment. It is indeed a rate over the reported percentage and we appreciate and accept your proposal considering a more careful and thoughtful approach, adopted. Therefore, we added a sentence in the main manuscript body (section: Discussion, green-highlighted), considering this aspect and we also considered the proposed references.

We also thank you very much for the second point of your review: After a thorough review of our tumor registry, we did not see any significant differences in the colorectal cases before and after certification:

  1. colon carcinoma

median (5 years before certification): 38 cases vs. 44 cases (5 years after certification) (p=0.113)

  1. rectal carcinoma

median (3 years before certification): 23 cases vs. 28 cases (3 years after certification)

(p=0.267)

Median (5 years before certification) 21 cases vs. 24 cases (5 years after certification) (p=0.285). The centers kept almost the same amount of cases treated, both for patients with colon cancer and for them with rectal cancer. Considering the last aspect, we firmly believe that a volume of 30-50 cases for colon cancer and a volume between 20 and 30 for rectal cancer (as proposed by the German Cancer Society) are the quantitative benchmarks for a well-functioning Certified Colorectal Cancer Center. In our honest opinion the qualitative aspects of a certification process (such as need for multimodal treatment, need for respecting and following the guidelines and the current state-of-the-art, individualized treatment options, need for interdisciplinary tumorboards etc.) are -together with stable volume of cases- the essential aspects for improvement. Thank you for your excellent remark. We added a paragraph in our revised manuscript (Section: Conclusion), pinpointing the importance of the mentioned aspects.

We thank you for the valuable comments, we accept the concerns of the reviewers, we revised the manuscript, and all corrections of the second review round have been green-highlighted.

All already performed corrections for the first review-round remained red-highlighted.
We thank you once again for consideration of our manuscript in light of our response to the reviewer`s comments. 

We very much hope that you will accept the revised manuscript for publication, as we believe we have performed all proposed revisions. All revisions and corrections in the manuscript are red- (first review round) and green-(second review round)-highlighted.

Sincerely yours,

Aristotelis Perrakis, Mihailo Andric on behalf of the co-authors.

Round 3

Reviewer 1 Report

Comments and Suggestions for Authors

The headline of your article states that data from 13 hospitals were analysed, whereas the article describes and calculates data from 26 hospitals. I do not understand the objectives of the publication anyway and the conclusions are very abstract. The article is overloaded.